# Investigation of DNA Hybridization on Nano-Structured Plasmonic Surfaces for Identifying Nasopharyngeal Viruses

**DOI:** 10.3390/bioengineering10101189

**Published:** 2023-10-13

**Authors:** Shao-Sian Li, Yi-Jung Lu, Ray Chang, Ming-Han Tsai, Jo-Ning Hung, Wei-Hung Chen, Yu-Jui Fan, Pei-Kuen Wei, Horn-Jiunn Sheen

**Affiliations:** 1Department of Materials and Mineral Resources, National Taipei University of Technology, Taipei 10608, Taiwan; ssli@mail.ntut.edu.tw; 2Division of Family and Operative Dentistry, Department of Dentistry, Taipei Medical University Hospital, Taipei 11031, Taiwan; yi_jung2002@yahoo.com.tw; 3Institute of Applied Mechanics, National Taiwan University, No. 1, Section 4, Roosevelt Rd, Taipei 10617, Taiwan; d11543004@ntu.edu.tw (R.C.); r09543071@ntu.edu.tw (W.-H.C.); 4Institute of Microbiology & Immunology, National Yang Ming Chiao Tung University, No. 155, Section 2, Linong St., Beitou District, Taipei 11221, Taiwan; m.tsai@gm.ym.edu.tw (M.-H.T.); cheeseerer0707@gm.ym.edu.tw (J.-N.H.); 5School of Biomedical Engineering, College of Biomedical Engineering, Taipei Medical University, Taipei 11031, Taiwan; 6Research Center for Applied Sciences, Academia Sinica, 128 Academia Road, Section 2, Nankang, Taipei 11529, Taiwan

**Keywords:** human herpesvirus 4, SARS-CoV-2, polymerase chain reaction, nanoslit-based plasmonics, label-free, duplex PCR

## Abstract

Recently, studies have revealed that human herpesvirus 4 (HHV-4), also known as the Epstein–Barr virus, might be associated with the severity of severe acute respiratory syndrome coronavirus 2 (SARS-CoV-2). Compared to SARS-CoV-2 infection alone, patients coinfected with SARS-CoV-2 and HHV-4 had higher risks of fever, inflammation, and even death, thus, confirming that HHV-4/SARS-CoV-2 coinfection in patients could benefit from clinical investigation. Although several intelligent devices can simultaneously discern multiple genes related to SARS-CoV-2, most operate via label-based detection, which restricts them from directly measuring the product. In this study, we developed a device that can replicate and detect SARS-CoV-2 and HHV-4 DNA. This device can conduct a duplex polymerase chain reaction (PCR) in a microfluidic channel and detect replicates in a non-labeled manner through a plasmonic-based sensor. Compared to traditional instruments, this device can reduce the required PCR time by 55% while yielding a similar amount of amplicon. Moreover, our device’s limit of detection (LOD) reached 100 fg/mL, while prior non-labeled sensors for SARS-CoV-2 detection were in the range of ng/mL to pg/mL. Furthermore, the device can detect desired genes by extracting cells artificially infected with HHV-4/SARS-CoV-2. We expect that this device will be able to help verify HHV-4/SARS-CoV-2 coinfected patients and assist in the evaluation of practical treatment approaches.

## 1. Introduction

The coronavirus disease 2019 (COVID-19), caused by severe acute respiratory syndrome coronavirus 2 (SARS-CoV-2), has rapidly spread globally. Many medical professionals use polymerase chain reaction (PCR) techniques to identify viral infections. With this technique, particular DNA fragments can be replicated, and fluorescence detection can be used to track the growth of amplicons. Despite their extensive use, conventional methods have drawbacks, such as high apparatus costs and laborious processes including labeling steps and heating cycles. Therefore, a solution to these issues was provided by the development of microelectromechanical systems (MEMSs) [1].

MEMS methods enable the downsizing of PCR processes by integrating multiple components onto a single chip [2,3]. Working fluid in a small (usually sub-millimeter) microfluidic channel can quickly heat up or cool down efficiently. Based on various research objectives, many microchannel designs have been used in PCR microfluidic devices [4,5]. For instance, target genes can be amplified by heating cycles. Felbel et al. reported a continuous-flow device that could run 35 PCR cycles (at 50 and 94 °C) via serpentine microchannels and different temperature zones [6]. A digital PCR system introduced by Mao et al. could distribute the target DNA into droplets and amplify the DNA in each droplet by 40 cycles of PCR (at 60 and 95 °C) [7].

After the reaction, the amplicon needs to be further analyzed. Current methods for biomolecular detection can be roughly divided into label-dependent and label-free approaches [8,9]. In label-dependent methods, foreign molecules, including fluorescent probes, radioisotopes, and nanoparticles, bind to the molecule of interest to determine its presence. Although these methods have high specificity, the labeling process is time-consuming, and samples occasionally lose their functionality after the analytical process. In contrast, label-free techniques can directly monitor target molecules, which utilize the molecular refractive index, weight, and charge. The surface plasmon resonance (SPR) technique is a label-free approach that can be configured in several ways and is compatible with various detection systems [10,11]. For example, Nguyen et al. incorporated an SPR fiber sensor with microfluidic PCR for pathogenic bacterium detection [12]. To quantify tumor-derived exosomes, Zhu et al. combined an SPR coupling prism with a microarray [13]. Due to their sensitivity and versatility, SPR tools have been extensively employed since the COVID-19 outbreak to identify SARS-CoV-2-related analytes, including the human immunoglobulin G (IgG), anti-spike protein, and viral single-stranded RNA and DNA sequences [14]. For instance, Cennamo et al. demonstrated that SPR optical fiber coupled with aptamer [15] for recognition of SARS-CoV-2 spike protein.

Recently, studies have revealed that the severity of COVID-19 may be correlated with the presence of human herpesvirus 4 (HHV-4). Patients who are coinfected with SARS-CoV-2 and HHV-4 have a greater risk of fever, inflammation [16], and even mortality [17,18] than those with SARS-CoV-2 infection alone. Therefore, prompt confirmation of HHV-4/SARS-CoV-2-coinfected individuals can aid clinical decisions for additional treatments. Although many devices have been fabricated to detect HHV-4 [19,20] or SARS-CoV-2 [14], a device that can simultaneously detect both has not been reported to our best knowledge. Therefore, we have now developed a device based on our preceding work [20] to address this issue. Previously, we reported an all-in-one chip composed of a microfluidic PCR and a nanoslit-based SPR for detecting the DNA sequence of latent membrane protein 1 (LMP1), a biomarker in various HHV-4-associated diseases [20]. Herein, we have optimized several parameters to enhance the device’s performance. In the fabrication process, solvent-based bonding was applied to join the acrylic sheets instead of using heat-resistant, double-sided tape, which enhanced the device’s long-term durability. Moreover, the microchannel parameter, PCR conditions, and primer design were modified to boost the device’s performance from single detection to duplex examination. The effects of lengths of targeting sequences during sensing was also investigated.

## 2. Materials and Methods

### 2.1. Integration of the Microfluidic Device

The SPR chip was produced via the previous work [20] (Figure 1). Briefly, the nanoslit structure of the mold, characterized by a 470 nm spacing and a 100 nm width, was replicated onto a polycarbonate film by hot-embossing nanoimprinting lithography (165 °C under 900 kPa), followed via old deposition through the sputtering system. For microfluidic PCR, a laser-scribing acrylic sheet with a defined structure was joined to another flat acrylic sheet through a solvent mixture of 20% acetone and 80% ethanol under uniform pressure (490 kPa) and temperature (60 °C for 3 min) [21]. Then, the chip was adhered to the microchannel via heat-resistant double-sided tape. Finally, the DNA probe was capped onto the chip in the device through the thiol-gold reaction and electrostatic adsorption [20]. Specifically, a 100 μg/mL cysteamine solution was fed into the device and kept for two hours. Then, the device was washed with DI water and incubated with probe solution for 10 min to achieve the modification.

### 2.2. The Operation of the Device

After introducing the DNA solution into the device, the target DNA is subjected to a duplex polymerase chain reaction (PCR) within a designated microfluidic channel. Subsequently, the resultant replicates are identified using a label-free approach via a plasmonic-based sensor.

For DNA amplification, two heating blocks were adjusted to 95 and 50 °C to carry out 30 loop thermal cycles in the microfluidic PCR. Meanwhile, for DNA detection, the space between the blocks kept the temperature near 65 °C, which is close to the probe’s melting temperature (T_m_), enhancing the sensitivity of the SPR sensor. The syringe pump maintains a consistent flow rate in the device.

For the amplicon detection, an optical system was set up to measure the resonant spectrum from nanoslit SPR sensors. Broadband white light was polarized in the transverse magnetic (TM) direction and then focused on the SPR sensor via a pinhole. After passing the SPR sensor, a fiber lens gathers the transmitted light into an optical fiber and then gauges it with a spectrometer (B&W Tek, Newark, DE, USA).

The calculation of red-shift value has been described in our previous work [20]. In particular, the resonant wavelength (*λ*) in the TM direction can be described as
*λ* = *a* · *d_p_*;
where *d_p_* is the period of the SPR chip and *a* is the water refractive index (*a* = 1.33). For spectrum analysis, the spectral centroid approach was utilized to measure the red-shift. The red-shift value was determined by performing a subtraction operation on the average wavelengths seen prior to and subsequent to the binding of DNA to the nanoslit. The experiment involved recording and graphing the resonant wavelength at different time intervals to analyze specific DNA binding stages.

### 2.3. PCR Analysis

*COVID-19 N-gene* (148 bp) and *LMP1* (286 bp) DNA primers and probes were designed according to previous studies [20,22]. Details of the primer and probes are as described in Appendix A. The PCR solution (EconoTaq^®^ PLUS GREEN 2X Master Mix) was purchased from Lucigen (Middleton, WI, USA). The solution composition for the PCR, duplex PCR, and machine conditions are described in Appendix A, Appendix A, and Appendix A, respectively.

### 2.4. Cells

HEK293 cells were derived from human embryonic kidney cells (ATCC: CRL-1573, American Type Culture Collection, Manassas, VA, USA). HEK293 cells were stably transfected with recombinant HHV-4-BACmid of the M81 strain (293/HHV-4) which was a gift from Prof. Henri-Jacques Delecluse (German Cancer Research Center, Heidelberg, Germany) and maintained in medium containing 100 μg/mL hygromycin (Invitrogen, Carlsbad, CA, USA). All cells used were maintained in RPMI medium (Invitrogen, Carlsbad, CA, USA) supplemented with 10% fetal bovine serum (FBS, HyClone, Cytiva, Marlborough, MA, USA) in a humidified cell culture incubator at 37 °C.

### 2.5. Plasmids

pLAS2 is a lentiviral vector with a CMV promoter, and a gene cassette carrying a puromycin-resistant gene was obtained from the RNAi core of Academia Sinica (Taipei, Taiwan). The *N-gene* of SARS-CoV-2 was directly synthesized and served as template for further experimentation, and the full sequence is given in Appendix A. We inserted the *N-gene* of SARS-CoV-2 (pLAS2-SARS-CoV-2-N) after the CMV promoter of the pLAS2 lentiviral vector (pLAS2-SARS-CoV-2-N) via PCR cloning with primers 5′-TTTGCTAGCCACATTGGCACCCGCAATC-3′ and 5′-AAAGATATCAGCTTGTGTTACATTGTATG-3′. The pLAS2 plasmid carrying a green fluorescent protein (*GFP*) gene after the CMV promoter was also constructed (pLAS2-GFP) and served as a control in our experiments.

### 2.6. Production of the Lentivirus and Cell Infection

VSV-G-based lentiviruses were produced by co-transfecting HEK293T cells with pCMVR8.91, pLAS2-based lentiviral plasmids pLAS2-SARS-CoV-2-N and pLAS2-GFP, and plasmids encoding either SARS-CoV-2 S or VSV-G at ratios of 6.25: 6.5: 1.1 using PEI transfection (PEI MAX, M.W. 40,000, Polysciences 24765-1, Warrington, PA, USA) following the manufacturer’s instructions. Supernatants were collected at 60 h post-transfection, passed through a 0.45-μm filter, and applied for cell transduction with 8 μg/mL hexadimethrine bromide (H9268, Merck, Darmstadt, Germany). Culture supernatants were replenished with fresh medium containing 1 μg/mL puromycin (Invitrogen, Carlsbad, CA, USA) for selecting cells successfully transduced with the indicated lentiviruses.

### 2.7. DNA Extraction

The extraction of DNA from cells was performed using a commercially available kit (AxyPrep Multisource Genomic DNA Miniprep Kit, 50 prep, Corning, NY, USA). All procedures were executed following the guidelines provided by the manufacturer. After extraction, the DNA concentration and purity of each sample were evaluated via NanoDrop (Thermo Scientific, Waltham, MA, USA).

### 2.8. RNA Extraction and Reverse Transcription

RNA was extracted from cells using a commercial kit (Total RNA Extraction Miniprep System, 50 prep, Viogene, Sunnyvale, CA, USA). The procedures adhered to the norms established by the manufacturer. After RNA extraction, samples were immediately turned into complementary (c)DNA via a reverse transcription process guided by the manufacturer’s protocols (TOOLSQuant II Fast RT Kit, BIOTOOLS Co., Ltd., Taipei, Taiwan). The cDNA concentration and purity of each sample were also examined via NanoDrop.

## 3. Results

### 3.1. Temperature Stability and Sensitivity of the Microfluidic Device

To determine the thermal stability, a thermographic camera was used to record the temperature distribution of the device (Appendix A). The results demonstrated that a homogeneous temperature distribution was observed around 10 min. On the other hand, the sensitivity of the SPR sensor could be verified from the slope (*R*^2^ = 0.996, Appendix A), which was 450.55 nm/RIU, close to the 470 nm interval for which the nanoslit was designed.

### 3.2. Optimal Flow Rate for Detection Using the Device

The PCR microchannel flow rate was optimized to effectively detect target DNA according to the results from gel electrophoresis and the red-shift of SPR (Figure 2). First, a PCR solution containing *LMP1* or the *COVID-19 N-gene* was pumped into the microfluidic PCR at various flow rates (1, 2, 3, 4, 5, 6, and 7 μL/min). Consequently, under flow rates ranging from 2 to 6 μL/min, distinct bands were seen at about 150 bp (*COVID-19 N-gene*) and 300 bp (*LMP1*) (Figure 2A), indicating the effective replication of DNA on the device under these specific flow circumstances. In contrast, the band disappeared with a flow rate of 1 or 7 μL/min, indicating that the flow rate was too slow or fast to produce detectable product concentration. On the other hand, despite the band being observed at a flow rate of 1 μL/min, this condition required a longer reaction time and had lower specificity. So, a flow rate of 1 μL/min was not further considered.

SPR results (Figure 2B) showed that the *COVID-19 N-gene*’s amplicon produced a greater red-shift at flow rates of 4 (4.93) and 5 μL/min (4.81) than at other flow rates. For *LMP1*, the replicate exhibited the highest red-shift at a flow rate of 5 μL/min (2.61) compared to the others. Although he *COVID-19 N-gene* displayed the highest red-shift at 4 μL/min, *LMP1* did not exhibit a remarkable red-shift (2.02) at this flow condition. Additionally, a flow rate of 4 μL/min required more reaction time (nearly 8 min) than that at 5 μL/min. Thus, a flow rate of 5 μL/min was chosen for the following experiments.

### 3.3. The Limit of Detection (LOD) of the Device

The device’s LOD was evaluated through serial dilutions, reducing the DNA concentrations from 10^−5^ to 10^−14^ g/mL (Figure 3). PCR solution without DNA served as the control group. 

For the *COVID-19 N-gene*, the traditional PCR replicate’s red-shift appeared linearly in the 10^−8^ to 10^−13^ g/mL, with an LOD at 10^−13^ g/mL (Figure 3A). Specifically, it was found that for every 10-fold increase in DNA concentration, the red-shift increased by 1.44 nm (R^2^ = 0.993). In contrast to conventional PCR, the amplicon in the device exhibited a linear red-shift within 10^−5^ to 10^−10^ g/mL, with an LOD at 10^−10^ g/mL (Figure 3C). The DNA concentration increased 10-fold in this range as the red-shift grew by 0.84 nm (*R*^2^ = 0.998). We suggest that this difference could be attributed to the traditional PCR machine performing *COVID-19 N-gene* amplification better than the device. At the same initial concentration, the red-shift from the traditional PCR replicate was 8 nm, while that of the device was 5 nm. 

On the other hand, the result of *LMP1* showed that the traditional PCR (*R*^2^ = 0.993, Figure 3B) and the device (*R*^2^ = 0.991, Figure 3D) exhibited linear regressions in the 10^−5^ to 10^−12^ g/mL with LOD at 10^−12^ g/mL. Although the above results show that the traditional instrument had better PCR performance, the device can remarkably shorten the reaction time. To perform the 30 cycles of amplification, the traditional PCR required 90 min, while the device only took about 40 min. Most importantly, amplicons from both methods displayed similar SPR red-shifts. In the linear concentration range, differences in the replicate’s red-shift between the traditional PCR and the device were lower than 3 and 1 for the *COVID-19 N-gene* and *LMP1*, respectively.

### 3.4. Specificity of the Device

The PCR solution, with the target or non-target gene, was introduced into the device to validate its ability to detect specific DNA sequences (Figure 4). For the solution containing the *COVID-19 N-gene*, the *COVID-19 N-gene* probe-modifying device demonstrated a red-shift within a linear range of 10^−13^ to 10^−8^ g/mL (*R*^2^ = 0.998, Figure 4A). However, no notable red-shift was detected using the device modified with the *LMP1* probe. On the other hand, the *LMP1* solution showed a linear trend from 10^−12^ to 10^−5^ g/mL (*R*^2^ = 0.991) exclusively with the *LMP1* probe-modifying device. These results indicated that the device could discern the gene after modifying the probe.

### 3.5. Duplex PCR of the Device

Duplex PCR is the simultaneous amplification of two targets in a single reaction chamber, with a separate pair of primers for each target. Herein, the capability of the device to conduct duplex PCR was investigated (Figure 5). First, a PCR solution containing two genes was pumped into the device for the duplex PCR, which was further analyzed via gel electrophoresis to optimize the composition of the PCR solution. Then, the red-shift of the optimized duplex PCR product constructed by the traditional machine or the device was examined.

Gel electrophoresis identified condition 6 (0.2 μM *COVID-19 N-gene* and 0.2 μM *LMP1*) as having the optimal band position and intensity; thus, it was selected for the duplex PCR solution (Figure 5A). Then, the PCR solution was amplified using a conventional machine (Figure 5B) or the device (Figure 5C). As with earlier findings (Figure 3), the traditional machine produced higher red-shifts than the device, albeit the difference was minor (<5, Figure 5). The linear trend (R^2^ ≥ 0.99, Figure 5B,C) from the duplex PCR was parallel to single-specimen results (Figure 3 and Figure 4), confirming that the device can accurately perform duplex PCR and detect the desired gene in the mixture.

### 3.6. Detection of LMP1/COVID-19 N-Gene in Artificial HHV-4/SARS-CoV-2-Modified Cells

Further, to confirm that the device can replicate and detect *LMP1*/*COVID-19 N-gene* in cell-extracted DNA/cDNA, HEK 293 cells containing recombinant HHV-4 strain M81 transduced with the *COVID-19 N-gene* were used to mimic HHV-4/SARS-CoV-2-infected cells. Although SARS-CoV-2 is an RNA virus, the isolated RNA can be reverse-transcribed into cDNA and then analyzed by the device. The results revealed that artificially HHV-4/SARS-CoV-2-infected cells (Lanes 2 and 3) had remarkable *LMP1* and *COVID-19 N-gene* PCR bands (Figure 6A) and red-shifts (Figure 6B). It should be noted that the specimen with *COVID-19 N-gene* cDNA/DNA (Lane 3) exhibited marked *COVID-19 N-gene* expression compared to the specimen solely with *COVID-19 N-gene* DNA (Lane 2). On the other hand, artificially HHV-4- (Lane 1) or SARS-CoV-2-infected (Lane 4) cells, respectively exhibited the *COVID-19 N-gene* blue-shift or *LMP1* red-shift (~0.13), both of which were small enough to be omitted. These results proved that the device could replicate and evaluate the extractions from HHV-4/SARS-CoV-2 infected cells in a duplex manner.

## 4. Discussion

This study revealed that the device could replicate and detect the HHV-4 and SARS-CoV-2 genes. Although this work expands on our earlier investigation [20], we modified the microchannel parameter, PCR conditions, and primer design to boost the device’s performance. Moreover, it should be noted that the SPR signal from the *COVID-19 N-gene* was stronger than that from *LMP1* (Figure 3, Figure 4 and Figure 5), which might could be attributed to the interaction of the probe and the target DNA. Research has reported that the degree of DNA’s self-entanglement in an aqueous solution is proportional to its length [23]. Additionally, DNA folding may interfere with the interaction of a biomolecule binding to specific sites on DNA [24]. Electrophoresis (Figure 2 and Figure 5) showed that the length of *LMP1* (~300 bp) was remarkably longer than the *COVID-19 N-gene* (~150 bp), which indicated that *LMP1* DNA was more likely to self-entangle. Furthermore, the probe-binding site of the target DNA is in the central region of the sequence (Appendix A). Thus, we suggest that *LMP1* would affix to the probe on the sensor surface with greater difficulty than the *COVID-19 N-gene*, resulting in a lower SPR signal.

On the other hand, several intelligent devices could simultaneously discern multiple genes related to SARS-CoV-2 [25,26,27], but most operated via label-based detection, making it hard to directly investigate the product. Although Sardar et al. reported a nanoplasmonic-based biosensor for label-free screening tests [28], their work focused on the design of material utilized in a 96-well plate instead of fabricating an integral microdevice. Herein, we prepared a device that can conduct a duplex PCR in a microfluidic channel and non-labeling detect replicates through a nanoslit sensor. The low-cost device can easily be created via laser scribing and hot-embossing nanoimprinting lithography. When comparing the device to traditional instrument, the device is capable of reducing the PCR time by 55% while yielding a similar amount of amplicon as the instrument (Figure 3 and Figure 5). Moreover, the sensitivity of prior non-labeled sensors for SARS-CoV-2 detection was in the ng/mL to pg/mL range [29,30], whereas our device’s LOD can reach 100 fg/mL (Figure 3A). Finally, we verified that the device could detect the desired genes from extraction of artificially HHV-4/SARS-CoV-2-infected cells (Figure 6). We will adjust the microchannel and optimize the PCR primer to enable the device to concomitantly detect numerous genes.

## 5. Conclusions

We have presented a plasmonic-based duplex biosensor to detect DNA from SARS-CoV-2 and HHV-4. When introduced into the device, cellular DNA undergoes simultaneous replication and recognition. Compared to traditional instruments, our device trims the PCR duration by 55% without compromising amplicon yield. Our findings also reveal that longer probe sequences generate a diminished SPR signal compared to shorter ones. Hence, our future research will delve into the interplay between sequence length and SPR signal strength, potentially guiding the design of future SPR-based instruments. We anticipate this device can assist in identifying patients with HHV-4/SARS-CoV-2 co-infections, supporting the evaluation of effective treatment strategies.

## Figures and Tables

**Figure 1 bioengineering-10-01189-f001:**
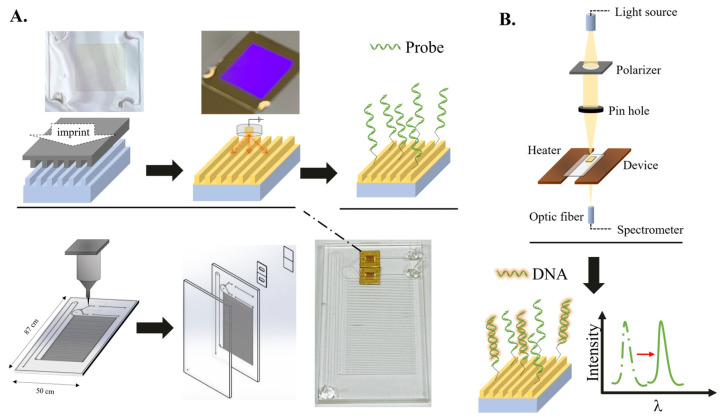
Overview of the detection system. (**A**) The device comprises a microchannel and a gold-capped nanoslit chip. The acrylic sheet substrate was trimmed via a laser and engraved with a microfluidic channel. Then, the imprinting nanoslit was adhered to the microchannel to create the device. After fabrication, the nanoslit was modified with the probe. For target detection, 95 and 50 °C for denaturation and annealing/extension were set on two sides, while the device’s center was sustained at 60 °C for detection by heat transfer. (**B**) The configuration of the system. An optical system was set up to measure the resonant spectrum from nanoslit SPR sensors.

**Figure 2 bioengineering-10-01189-f002:**
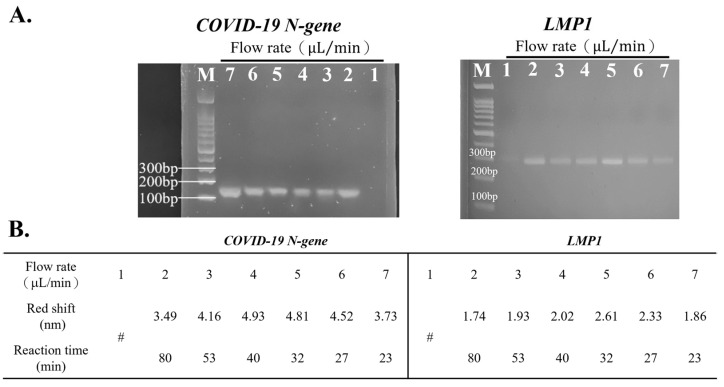
Optimal flow rate for the device’s detection. (**A**) Replicates produced by the device at varying flow rates were verified via gel electrophoresis. (**B**) The products’ red-shift produced with various flow rates. # The flow rate of 1 μL/min showed in gel was not considered due to the longer reaction time and lower specificity.

**Figure 3 bioengineering-10-01189-f003:**
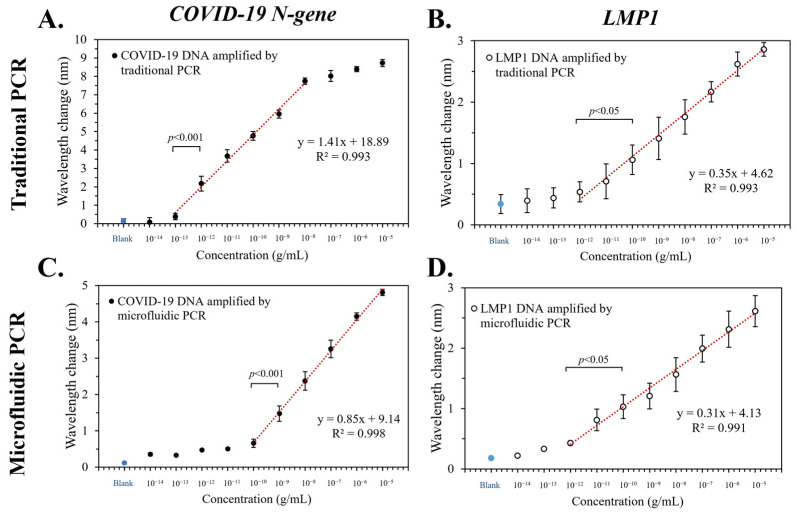
The limit of detection (LOD) of the device. Serial dilutions of genes amplified via a traditional machine (**A**,**B**) and the device (**C**,**D**) were analyzed. The control group consisted of a PCR solution that did not include DNA. Error bars denote the standard deviation (*n* = 3).

**Figure 4 bioengineering-10-01189-f004:**
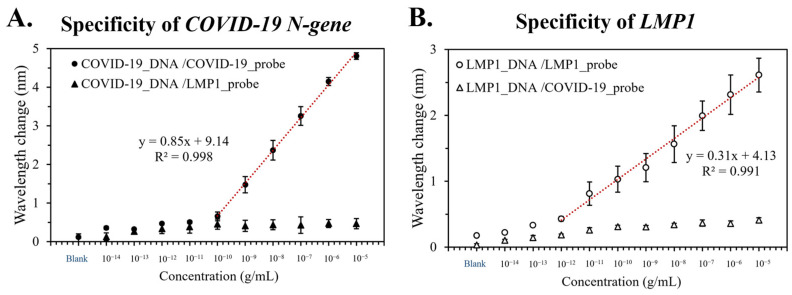
Specificity of the device. A PCR solution with the target or non-target gene was separately pumped into the device to verify the device’s specificity for the *COVID-19 N-gene* (**A**) or *LMP1* (**B**). Error bars denote the standard deviation (*n* = 3).

**Figure 5 bioengineering-10-01189-f005:**
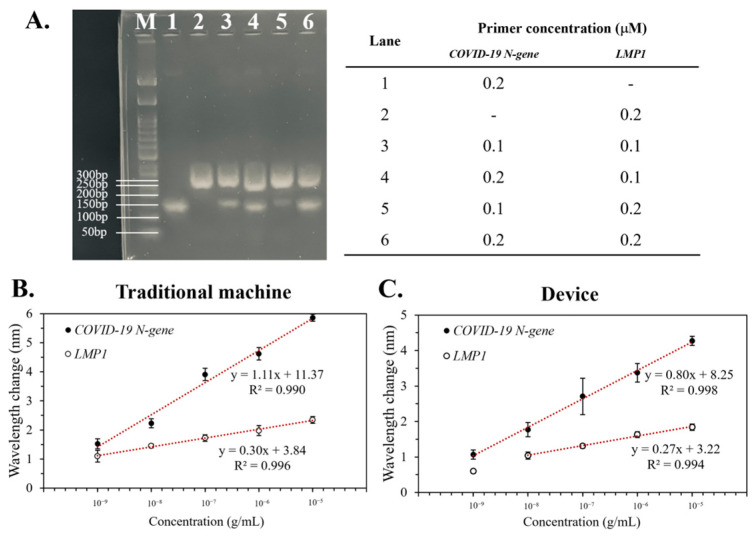
The duplex PCR in the device. Two target genes were simultaneously replicated and detected by the device. (**A**) The primer concentrations were optimized and confirmed through gel electrophoresis. The red-shift of the optimized duplex PCR product produced by the traditional machine (**B**) or the device (**C**) was examined. Error bars denote the standard deviation (*n* = 3).

**Figure 6 bioengineering-10-01189-f006:**
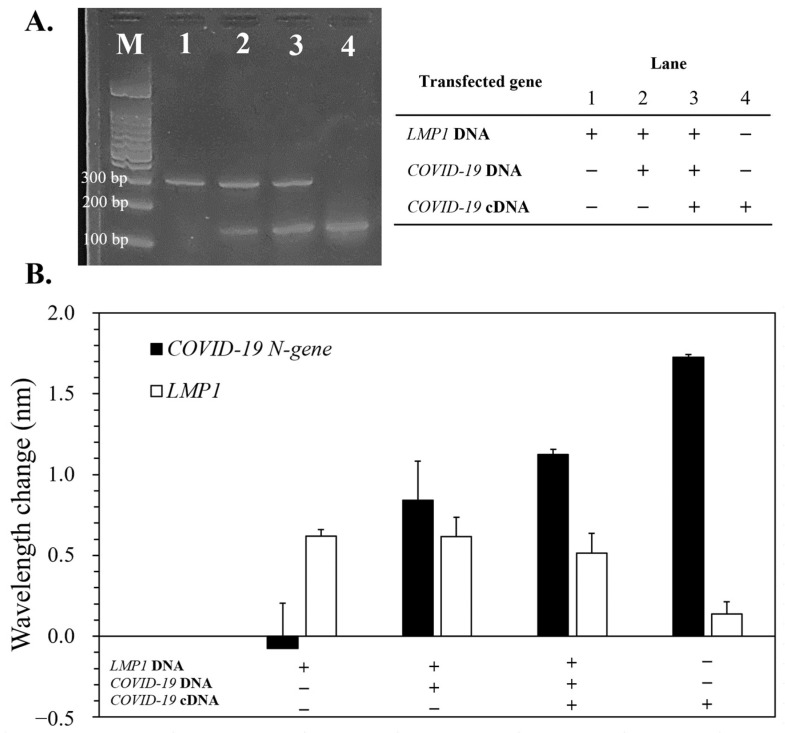
Detection of cell DNA extraction via the device. Cells transfected with HHV-4 (*LMP1*) DNA, SARS-CoV-2 (*COVID-19 N-gene*) DNA/cDNA were lysed and verified via (**A**) gel electrophoresis and (**B**) the device. The initial concentration of target DNA was 10 ng/μL. Error bars are used to represent the standard deviation (*n* = 3).

## Data Availability

Not applicable.

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
