# Peer review of "Investigation of DNA Hybridization on Nano-Structured Plasmonic Surfaces for Identifying Nasopharyngeal Viruses"

_bioengineering, 2023, doi:10.3390/bioengineering10101189_

Round 1

Reviewer 1 Report

Virus detection has important medical significance. This article establishes a small sensing device that can simultaneously detect two types of virus samples. This work is mainly based on existing research, only integrating two different sensing probes in the same device, without proposing targeted designs for multi sample analysis.

1. Some formatting errors, such as “where” on line 171 should be left aligned and roman, and “a” in parentheses should be italicized.

2. The experiment in Figure 2 should be repeated multiple times. For the amplified target fragment, the electrophoresis results are more pronounced at flow rate 2 than at flow rate 7, but why is the SPR signal weaker?

3. Section 3.3 only provides a linear range without a limit of detection.

4. In section 3.4, in order to verify the specificity of nucleic acid testing, the selected interfering substance is too single to reflect the interfering effect, and actual samples such as body fluids should be used.

Author Response

The response is attached.

Reviewer 2 Report

This work describes a device which can perform PCR in a shorter time using microfluidics. The target DNA is then, through the device, detected with SPR.

The work is a follow-up work to the author's previous device which worked with LMP1 only. This work detects 2 targets which informs better treatment options for the patient.

Introduction:

Authors clearly state the purpose and reason for their work- help better treatment options for patients having both illnesses.

Information supports advantages of current work over their own past work and work using other processing and detection methods.

This work: faster PCR, multi-targets, lower LOD than similar research for same targets. Label-free. 

Materials and Methods:

Authors adequately describe their methods for cell infection, PCR use, DNA extraction, and plasmids used.

Results:

Authors clearly explain reasoning for optimized flow-rate.

Authors discuss honestly the shortcomings of their device as compared to traditional methods (PCR) and present the advantages of their method logically.

The detection range and LOD for this work is within the relevant detection range.  Good choice of blanks and non-targets to show specificity of probes.

Paragraph in lines 252-261 could be clearer in the wording.

Paragraph lines 276-290 could be written in a smoother/flowy manner. Sentences seem to be unconnected, bullet-point style.

Section 3.6

It was not clear at first reading that signals for Figure 6 are from LMP1 signal and that the other signal is for both the Covid targets. Figure 6 B bars graph needs labeled y-axis. 

Paragraph in lines 252-261 could be clearer in the wording.

Paragraph lines 276-290 could be written in a smoother/flowy manner. Sentences seem to be unconnected, bullet-point style.

Line 324- delete "might"

Author Response

The response is attached.

Reviewer 3 Report

Dear Authors, 

The submitted MS ´Investigation of DNA Hybridization on Nano-structured Plasmonic Surfaces for Identifying Nasopharyngeal Viruses.´

The MS is well-written and organized. it explains various methods of cell, PCR, and DNA/RNA extraction. 

Since it is an extension of the previously published work, it strengthens the research publication.  

However, it is suggested that a comparison with a commercial device would be good if it is discussed and validated. 

The conclusion section is very short please elaborate on the findings and project the future direction of the research. 

Author Response

The response is attached.

Reviewer 4 Report

The work is interesting. However, there are few queries to improve the manuscript.

1.  The figure of the device is not clear. Need more clarity about the chip.  Please provide more clear images of the device (figure1).

2. Also explain the working methodology of the device. 

3.  There is no comparison of the work with previous literature. 

Need minor correction 

Author Response

The response is attached.

Round 2

Reviewer 1 Report

The paper has met the requirements for publication.